# Results from a Nationwide Evaluation Study of Labor Market-Integrative Health Promotion for the Unemployed: Impact of the JOBS Program Germany

**DOI:** 10.3390/ijerph20196835

**Published:** 2023-09-26

**Authors:** Alfons Hollederer, Heiko J. Jahn

**Affiliations:** Section of Theory and Empirics of Health, Department of Social Work and Social Welfare, The Faculty of Human Sciences, University of Kassel, D-34109 Kassel, Germany; heiko.jahn@uni-kassel.de

**Keywords:** unemployment, health promotion, prevention, mental health, JOBS program, intervention, randomized controlled trial, Germany

## Abstract

Compared to the employed, the unemployed suffer from poorer health, especially in terms of mental health. At the same time, health promotion rarely reaches unemployed people. The “JOBS Program” is an intervention to promote health and labor market integration and has shown positive effects in the USA and Finland. In this confirmatory study, we investigated whether the JOBS Program achieves similar effects in Germany. We applied a randomized controlled trial to compare an intervention group (IVG) with a waiting control group (WCG) before (T0; N = 94) and shortly after (T1; n = 65) the intervention. Concerning our primary outcomes, the JOBS Program Germany was beneficial: Compared to the WCG, the regression estimated that the IVG had (1) a 2.736 scale point higher level of life satisfaction (*p* = 0.049), (2) a 0.337 scale point higher level of general health (*p* = 0.025), and (3) a 14.524 scale point higher level of mental well-being (*p* = 0.004). Although not statistically significant, job search-specific self-efficacy also appeared to be positively associated with the intervention. This study provides evidence of the effectiveness of JOBS Program on the abovementioned outcomes, including for older and long-term unemployed people, supporting the benefits of regular implementation of this program for a wide range of unemployed people in Germany.

## 1. Introduction

In August 2023, the unemployment rate in Germany was 5.7%, with 2,695,827 unemployed people. Of these, 929,000 were long-term unemployed (34.5%) [1]. From a health sciences point of view, much is known about the health disadvantages that the unemployed experience. Compared to the employed, they suffer more frequently from somatic health problems like cardiovascular or musculoskeletal disorders [2,3] and are at higher risk of mortality due to diseases [4,5] and suicide [6,7,8]. In addition, unemployment turned out to be a risk factor for substance-use disorders [9]. However, mental health impairments, including depressive symptoms, are the main consequence of unemployment [10,11,12,13,14,15,16,17] and the association between unemployment and mental health impairments is empirically well proven [10,18]. Different psychological theories and models, such as stress and coping models or stigma theory, have already been transferred to and applied in research on unemployment and health [19]. In addition, there are theoretical approaches that have been developed specifically within social psychological research on unemployment and its consequences for mental health. Three particularly relevant theories and models, which are intended to explain which consequences of unemployment have a negative impact on mental health, should be briefly mentioned here as examples.

First, the “Latent Deprivation Model” according to Jahoda (1983) [20] primarily emphasizes—in addition to the stress caused by the loss of income—the lack of important (latent) functions of work, which can lead to impaired mental health. The resulting issues include a lack of social contacts, of a daily structure, of participation in joint efforts or goal achievement and of the necessary regular activity. In addition, Jahoda (1983) [20] postulated that an unemployed person may suffer from a change in social identity or social status.

Second, the “Vitamin Model” created by Peter Warr lists from 9 to 12 “principal aspects of any environments” ([21] p. 87) which also occur in the working environment, and whose presence or absence and extent can influence mental health [21,22,23]. There is some overlap with Jahoda’s latent functions of work (for example, social contacts and social status), but also other aspects such as the clarity/transparency of the (work) environment, equity within an organization, or career prospects.

Third, the “Agency Restriction Theory” described by David Fryer (1986) [24] focuses on the impaired autonomy of action of people who find themselves unemployed. According to the “Agency Restriction Theory” [24], people strive to be able to act independently and in this context, Fryer focuses on income that is intended to secure a living. Fryer’s theory assumes—in contrast to Jahoda [20]—that the financial restrictions resulting from unemployment are the major influencing factor in the impairment of mental health, because material poverty prevents people from independently shaping their lives according to their own ideas [19].

Internationally, there is a whole range of interventions for the unemployed that address mental health, with the aim of facilitating labor market integration [18,25,26,27,28]. One internationally evaluated intervention for the unemployed is the “JOBS Program”. This intervention was developed in the 1980s by psychologists at the Michigan Prevention Research Center of the Institute for Social Research, University of Michigan, USA, and aims primarily to increase personal resources such as self-confidence, self-esteem, and, first and foremost, self-efficacy [29,30,31]. The JOBS Program has been applied in different countries and has demonstrated positive results with regard to improving mental health and/or labor market integration, for example, in the United States [30,32,33,34], in Finland [35,36], Israel [37], Ireland [38], the Netherlands [39], China [40], and South Africa [41].

Since 2020, the JOBS Program has also been offered in Germany as “JOBS Program Germany” as a pilot project within the overarching large-scale program called “Linking of Employment and Health Promotion in the Community Setting”. Hereby the Federal Centre for Health Education (BZgA) and the National Association of Statutory Health Insurance Funds (GKV-Spitzenverband) cooperated with the German Federal Employment Agency.

It is not self-evident that an already internationally positively evaluated intervention shows the same effects under the conditions in another country because the practical implementation and its outcomes can depend on the country-specific labor market structure and the health and social security systems, respectively. Therefore, the pilot implementation of the JOBS Program in Germany was scientifically accompanied and evaluated by the University of Kassel to examine whether it had similar effects in Germany. In this sense, this study was confirmatory in nature and was intended to follow the methods and objectives of the abovementioned evaluation studies that have successfully been conducted by Vinokur et al. (2000) [42] in the United States and by Vuori et al. (2002) [35] in Finland.

This confirmatory study examined the effects of the JOBS Program on the job search-specific self-efficacy and on health-related outcomes such as life satisfaction and depressive symptoms among the unemployed in Germany. Of secondary interest were moderating factors like the duration of unemployment and the baseline level of depressive symptoms. Despite numerous difficulties in the participant recruitment and the practical training implementation caused by the COVID-19 pandemic-related lockdowns and other infection control measures, seven JOBS trainings took place, with 94 unemployed individuals participating in the first interview (T0, before the training), and we are able to report relevant results on the intervention effects related to job search-specific self-efficacy, life satisfaction, and general and mental health outcomes.

Concerning these outcomes, we hypothesized that the intervention group would report (1) higher levels of job search-specific self-efficacy and (2) life satisfaction, (3) to assess their general health status more favorably, and (4) to have lower levels of depressive symptoms compared to the waiting control group.

Following the study design and methods of the studies mentioned above, we also examined previously identified effect moderations. We hypothesized (5) that the level of depressive symptoms before the JOBS Program training (hereafter “JOBS training”) will moderate the effect of the intervention on mental health, in the way that the training would show stronger effects among those who suffered more from depressive symptoms. We also hypothesized (6) that the duration of unemployment before the JOBS training will also moderate the training effects on mental health.

## 2. Materials and Methods

### 2.1. Trial Design, Recruitment and Data Collection

This confirmatory study was designed as a multi-center, non-blinded, two-armed, parallel-group, randomized controlled trial (see details on the methodology in [43]) and the study design was based on two previous studies from the USA and Finland [35,42].

As part of the pilot launch of the JOBS Program in Germany, volunteer employment agencies were asked to inform their clients about the intervention and to invite them to voluntarily participate in the JOBS training and the evaluation study. If clients were interested, they were invited to an information event where they received detailed information about the training and the study. If participants gave their informed consent to participate in the study, they were called via telephone to conduct the first interview before the JOBS training (T0).

All interviews were conducted via Computer Assisted Telephone Interviews (CATI) with the software Voxco by the Institute for Social Sciences and Communication (SOKO; https://soko-institut.de/) on behalf of BZgA, accessed on 22 December 2022. During the interviews, the participants were randomly assigned to either the intervention group (IVG) or the waiting control group (WCG) (1:1 ratio) and invited to the JOBS training. After training, both groups were interviewed a second time as soon as possible, but within four weeks of training (T1). WCG participants were offered free participation in JOBS training after the study was completed.

Eligible study participants were adults who were registered unemployed with their local employment agency and were able to complete the CATIs independently in German. Due to occupational health and safety regulations during the COVID-19 pandemic, the employment agencies were not allowed to have personal contact with their customers for a large period of the recruitment, so they were faced with the challenge of inviting customers by phone or e-mail as well. The country-wide recruitment was expected to start in April 2021 and should continue until August 2021. Due to constraints imposed by COVID-19 pandemic-related infection control measures, the main study phase JOBS trainings took place between March and December 2022 (subsequent to a pretest that had been conducted to test all study processes).

### 2.2. Intervention

The JOBS Program uses elements of Social Learning based on Albert Bandura’s Social Cognitive Learning Theory [44,45] and Self-Efficacy Theory [46,47,48].

The JOBS Program training is designed as a multimodal workshop, usually lasting five days and from four to five hours for each session. Under the guidance of two certified trainers, participants develop and improve their practical job-search skills in small groups of 15 to 20 individuals. In terms of methods, the focus is on the following basic elements and group techniques:Job-search skills training.Active teaching and learning methods.Trained trainers for program delivery.Supportive learning environment.Preparation for setbacks.

Typical activities include job searching on social networking sites, compiling information for interviews, simulated job interviews, thinking in terms of employer perspectives, and evaluating job offers. Another essential component of the JOBS Program is the inoculation against—possibly discouraging—setbacks during the job-search. To be prepared against such setbacks, stress inoculation training is applied. The group anticipates potential difficulties in specific job-search situations and develops appropriate problem-solving strategies. All training content is taught using active teaching and learning methods. The participants’ prior knowledge and skills are identified and always incorporated into the various exercises. These are characterized by group discussions, brainstorming and role plays. For example, during the role plays, the participants take on an employer’s perspective to reflect their views and to develop respective strategies for application activities. Two further components accompany all activities of the entire training:Trainers provide continuous supportive feedback to participants and encourage appreciative, respectful interaction among the participants. They create an atmosphere of social support. Trainers also show empathy for participants’ concerns and feelings and encourage them to use appropriate coping strategies [29].Another training principle is the so-called “referent power”. The trainers strive to gain high esteem, trust and respect from the participants through competent teaching, self-revelation, the reduction of social distance and empathic support.

If both components can be successfully implemented, this appreciative and supportive training situation opens up better opportunities for the trainers to exert a positive influence on the participants’ self-efficacy expectations, on their self-esteem, and thus on their motivation to apply for a job [29,30].

A specification of the JOBS Program Germany is that the JOBS training sessions were free of charge. They lasted approximately 20 h (conducted over a period within one or two weeks) in groups of 8–15 participants and were led by two certified JOBS Program trainers [49]. One of the trainers should have been either qualified for adult education or professionally active in employment services. The other trainer had to be unemployed or should have at least unemployment experience. Both had to undergo specific training to be certified as JOBS Program trainers.

### 2.3. Predictor Variables and Outcome Measures

All predictor variables and outcomes were measured through a questionnaire developed by the research team at the University of Kassel. During the questionnaire-based CATI, data were collected on demographic characteristics, work and unemployment history, job-search intensity, reemployment, self-efficacy expectations, life satisfaction, and physical and mental health. Demographic characteristics were assessed using standard survey questions concerning age, gender, marital status, education, occupation, and length of unemployment. The level of education was determined by asking for the highest formal school-leaving qualification and the highest vocational qualification. For the analyses, this information was combined to construct dummy variables according to the International Standard Classification of Education (ISCED) [50,51], divided into three levels (low, medium, and high). The duration of unemployment was calculated as a continuous variable for years of unemployment. No dummy variable was constructed due to the lack of variance (only n = 11 [12%] participants were unemployed for less than 12 months).

The outcome measure job-search-specific self-efficacy was measured using a single four-point Likert-type item. The respondents were asked “Now that you think about the future, how likely do you think it is that you will get a job again?” and were able to select one of four predefined options (1 = “very unlikely”; 2 = “somewhat unlikely”; 3 = “somewhat likely”; 4 = “very likely”). The health-related outcomes were assessed based on (1) life satisfaction with a German version of the Satisfaction with Life Scale (SWLS) [52] and (2) participants’ self-evaluation of their general health status (“How is your health in general?”; five-point Likert-type item labeled from 1 = “very good”; 2 = “good”; 3 = “fair”; 4 = “bad”; 5 = “very bad”) [53]. Additionally, we examined (3) mental health (depressive symptoms) using the WHO-5 Well-Being Index [54,55]. The data were analyzed exclusively by the research team of the University of Kassel.

### 2.4. Statistical Analyses

All analyses have been conducted with the statistical software package SPSS Version 28.0. We treated Likert-type items as continuous variables. For scales, based on multiple items like the ones for life-satisfaction and depressive symptoms, we calculated additive scores according to the respective scale documentation.

We carried out standard descriptive analyses and—depending on the data measurement level, number of categories, distribution and cell counts—conducted χ^2^ tests or Fisher–Freeman–Halton exact tests for categorical variables as well as non-parametric Mann–Whitney-U-tests or t-tests for continuous variables to conduct group comparisons of continuous variables. This was carried out (1) to describe the sample (Table 1), (2) to examine the extent to which the randomization was successful, (3) to conduct a drop-out analysis, and (4) to identify significant group differences between the two groups at T0 and T1, particularly in terms of the focused outcome variables. We also aimed to identify significant changes in the outcomes from T0 to T1 within the IVG and the WCG using t-tests or Wilcoxon tests for paired samples. To identify multicollinearity, we created a correlation matrix for all variables studied. According to Field [56], a strong correlation was assumed with a (Spearman’s rho rank correlation) coefficient (r_s_) above 0.8.

We used ordinary least squares linear regression models adjusted for baseline values of the respective outcomes. We conducted sequential linear regression models that build upon each other. According to Urban & Mayerl [57], sequential regression analysis has an advantage: it can control for the dependence of the estimate of individual predictor effects on the estimated effects of other predictors in the model. In addition, the sequential procedure can easily identify both stable and unstable predictor effects, as well as those predictors that have a strong influence on the estimate for other predictor effects. In this way, it is also possible to investigate how strongly individual estimated predictor effects are influenced by the inclusion or exclusion of other predictors in the model. By using sequential regression, the different models can be compared in terms of the increase in the coefficient of determination of the regression, and this increase can also be tested for statistical significance using an F-test. The F-test tests the increase in the coefficient of determination that is achieved by adding additional predictor variables in an extended model. Thus, it can be observed whether the model fit is relevantly improved via the inclusion of additional predictor variables in the regression model and whether the addition of those predictors is thus statistically reasonable [57].

Model 1 (M1) included (1) the treatment indicator variable contrasting the IVG vs. the WCG in terms of the effect on the outcome and (2) the variable representing the baseline values of the respective outcome. The latter was carried out because it is expected that the baseline (T0) value of the focused outcome has a relevant effect on the dependent outcome variable (at T1). This is especially true if the comparison groups (IVG vs. WCG) differ regarding the baseline value. For this reason, it is recommended that baseline outcome values be included as a predictor in pre–post randomized controlled trials comparing the efficacy of two competing treatments (JOBS training vs. no intervention) [58,59]. Given the relevance of the baseline value of the outcome, it would not be meaningful to include only the treatment indicator variable in model 1. In model 2 (M2), demographic variables and variables representing the duration of unemployment and the level of depressive symptoms were also included. Those independent predictors were selected (1) on a theoretical basis or (2) if they have shown to be associated at a significance level < 0.2 with the outcome variable in different bi- and multivariate pretests. In the third model (M3), we further included interaction terms between our treatment indicator and the respective moderators to examine the abovementioned effect moderations (duration of unemployment, level of depressive symptoms at T0).

Except for the correlation analyses, in which pairwise analyses were performed (Table 2), all analyses were performed as listwise deletion (complete-case) analyses and according to the intention-to-treat principle (treating participants as if they belonged to the group to which they were originally (randomly) assigned, regardless of what treatment (if any) they received (in this case, the JOBS training) [60]). A *p*-value of less than 0.05 was considered statistically significant. Unless otherwise noted, the results of the two-sided significance test are reported.

## 3. Results

### 3.1. Participant Recruitment

Between March and December 2022, seven JOBS trainings were implemented. In total, 137 individuals signed the consent form and provided their contact data. Ninety-four participants completed the first interview and sixty-five completed the second interview, respectively (response rates 68.6 and 47.4%) (Figure 1).

### 3.2. Randomization and Dropout Analysis

To determine whether the statistical analyses were indeed performed according to a randomized controlled design, we compared the demographic and the outcome variables for statistically significant differences between the IVG and WCG at T0 (N = 94). There were only a few differences between the two groups at T0 in demographic variables, including duration of unemployment. The largest difference between the two groups was related to gender, as the proportion of men was greater in the IVG than in the WCG (58 vs. 50%) (Table 1).

**Table 1 ijerph-20-06835-t001:** Baseline (T0) characteristics of the intervention and waiting control group (N = 94).

	IVG *		WCG *		
	n = 50	%/Mean	n = 44	%/Mean	*p*-Value
Age (Years), Mean (SD)	50	44.6 (12.0)	44	44.8 (11.78)	0.964
Gender ^1^					
Male	29	58.0	22	50.0	
Female	21	42.0	22	50.0	0.437
Education					
Low Education	19	38.0	17	38.6	
Medium Education	27	54.0	23	52.3	
High Education	4	8.0	4	9.1	0.976
German citizenship					
Yes	44	88	39	88.6	
No	6	12	5	11.4	0.924
Duration of unemployment					
(Years), Mean (SD)	50	6.27 (6.38)	42	6.54 (5.84)	0.605

* A *p*-value of less than 0.05 was considered statistically significant. IVG: intervention group; WCG: waiting control group. SD: standard deviation. ^1^ In the survey, the question for gender provided the following categories: “male”, “female”, “diverse” (no respondent selected “diverse”).

For the outcome variables, there were two variables that differed somewhat strongly between the IVG and the WCG at T0: Compared to the WCG, participants in the IVG had higher scores on life satisfaction (19.38 vs. 17.68 points) and the WHO-5 Well-Being Index (depressive symptoms; 50.50 vs. 43.44 points). If one restricts this analysis only to the participants who were interviewed at both times T0 and T1 (n = 65), the same picture emerges (Table 3). However, for all variables reported here, we found no statistically significant differences between the two groups at T0, confirming that the integrity of randomization was assured.

In addition, we examined whether persons who participated in the two survey time points (T0, T1) differed from participants who participated in the first survey only (T0) (dropouts). Therefore, we compared data from these two groups regarding demographic and outcome variables at the time of the first survey (T0). The dropouts were slightly older on average (1.7 years). While the ratio of men to women was almost balanced in the group of those who had participated in the survey at both times (49.2 vs. 50.8%), the proportion of men in the group of dropouts was larger (65.5%). The educational level of dropouts was slightly lower than that of respondents (high/medium educational level: 55.2 vs. 64.4%), and there were no relevant differences concerning the respondents’ citizenship. Dropouts had spent, on average, 1.5 fewer years as unemployed and had slightly higher scores on job search-specific self-efficacy (0.28 points) and life satisfaction (0.35 points). The groups also showed little differences in terms of general health and well-being. The dropouts more frequently reported having good or very good general health than participants (41.4 vs. 40.0%) and scored slightly lower on the WHO-5 Well-Being Index (45.0 vs. 48.1 points). As these small differences suggest, for all variables reported here, none of the group comparisons were statistically significant. This indicates that there was no systematic bias in the results due to dropouts.

### 3.3. Participant Characteristics

The mean age of the participants (n = 94) was 44.7 years (standard deviation [SD] = 11.8). Slightly more than half of the participants were men (n = 51; 54.3%) and singles (n = 52; 55.3%). Thirty-six (38.3%) individuals had a low, 50 (53.2%) a medium, and 8 (8.5%) a high level of education, respectively. The vast majority (n = 83; 88.3%) had German citizenship and the mean duration of unemployment was 6.4 years (SD = 6.1) (Table 1).

### 3.4. Bivariate Correlation Analysis

Table 2 shows the correlations between all examined variables based on the information provided by participants interviewed at both time points T0 and T1 (max. n = 65). The associations between the intervention variable and the four outcome variables at T1 were negligible to weak, with the largest and statistically significant correlation coefficients of 0.31 for both life satisfaction and depressive symptoms, respectively. For all four outcome variables, statistically significant and plausible associations were identified between their T0- and T1-values, with moderate positive correlation coefficients ranging from 0.50 to 0.66. It is worth mentioning the associations between depressive symptoms and life satisfaction, with the strongest coefficient being observed for the positive significant correlation between depressive symptoms at T1 and life satisfaction at T1 (r_s_ = 0.64). Among the predictor variables, education was significantly correlated with age (r_s_ = 0.36) and the duration of unemployment was significantly correlated with gender (r_s_ = 0.52). Since the analyzed predictor variables did not show strong correlations with each other, there was no evidence of multicollinearity in the bivariate analysis.

**Table 2 ijerph-20-06835-t002:** Correlations matrix, means, standard deviations, and number of cases of predictor and outcome variables.

	Variable	1	2	3	4	5	6	7	8	9	10	11	12	13	14
1.	Intervention	–													
2.	Age (Years)	0.044	–												
3.	Gender	−0.078	−0.085	–											
4.	Education	0.090	0.363 **	−0.113	–										
5.	German Citizenship	0.066	0.066	−0.044	0.087	–									
6.	Duration of unemployment (Years)	−0.066	0.133	0.524 **	−0.096	0.005	–								
7.	Job search-specific self-efficacy, T0	−0.085	−0.240	−0.106	−0.033	0.034	−0.220	–							
8.	Job search-specific self-efficacy, T1	0.139	−0.210	−0.258 *	0.117	0.162	−0.282 *	0.501 **	–						
9.	Life satisfaction, T0	0.190	0.077	0.078	0.053	0.123	0.111	−0.054	−0.015	–					
10.	Life satisfaction, T1	0.312 *	0.087	−0.024	0.007	0.318 **	0.021	−0.139	−0.049	0.606 **	–				
11.	General health status, T0	0.084	−0.216	−0.153	0.082	0.046	−0.277 *	0.174	0.314 *	0.291 *	0.350 **	–			
12.	General health status, T1	0.172	−0.138	−0.173	0.236	0.207	−0.244	0.100	0.248	0.216	0.307 *	0.661 **	–		
13.	Depressive symptoms, T0	0.044	−0.006	−0.150	−0.020	0.260 *	0.033	0.289 *	0.192	0.368 **	0.422 **	0.327 **	0.350 **	–	
14.	Depressive symptoms, T1	0.305 *	−0.024	−0.163	0.056	0.327 **	−0.091	0.181	0.149	0.486 **	0.639 **	0.377 **	0.525 **	0.646 **	–
	Mean	0.52	44.15	0.51	1.74	0.89	6.85	2.79	2.89	18.46	20.11	3.28	3.14	48.13	53.69
	Standard Deviation	0.50	11.70	0.50	0.62	0.31	6.61	0.95	0.93	6.18	6.69	0.96	1.01	23.74	26.14
	Number of cases	65	65	65	65	65	64	61	61	65	65	65	65	63	64

* *p* < 0.05, ** *p* < 0.01. A *p*-value of less than 0.05 was considered statistically significant. Reported correlation coefficients are Spearman’s rank correlation coefficients. Results are based on the information provided by participants interviewed at both time points, T0 and T1 (max. n = 65).

### 3.5. IVG vs. WCG Differences at T0 and T1 and Outcome Changes from T0 to T1 in Both Groups

Table 3 shows the values for the outcomes in the IVG and the WCG for both survey time points T0 and T1, with the associated significance tests for differences between the groups at both time points. Table 3 also shows the extent to which the outcomes changed from T0 to T1 within each group, with the respective significance tests. All results are based on the information provided by participants interviewed at both time points, T0 and T1 (max. n = 65). There are no significant group differences in job search-specific self-efficacy at either T0 or T1. However, the increase in job search-specific self-efficacy from T0 to T1 is significant in the IVG, while the decrease in the WCG is not a significant change.

When assessing life satisfaction, positive developments can be identified. While the differences between the two groups were neither large nor significant at T0, it can be seen that this difference between IVG and WCG was larger and significant at T1. The increase in life satisfaction in the IVG from T0 to T1 was also significant. In contrast, there was virtually no change in life satisfaction within the WCG.

There were no significant results in terms of general health, either in terms of group differences at T0 or T1, or in terms of within-group changes from T0 to T1.

The clearest effect can be seen in the data on mental health. At T0, there were slight, non-significant differences between the two groups. However, since there was significantly better mental health in the IVG at T1 and little change in the WCG from T0 to T1, the differences between the two groups at T1 were significant and meaningful. This is also reflected in the fact that the improvement in mental health in IVG from T0 to T1 was confirmed with a rather small *p*-value (0.002).

**Table 3 ijerph-20-06835-t003:** Outcome values at T0 and T1 and changes from T0 to T1 by IVG and WCG.

	T0 ^1^			T1 ^2^			∆T0 to T1 ^3^
Outcome	IVG	WCG	*p*-Value	IVG	WCG	*p*-Value	*p*-Value
Job search-specific self-efficacy	2.72 (n = 32)	2.86 (n = 29)	0.511	3.03 (n = 33)	2.71 (n = 28)	0.280	IVG: 0.049 * (pairs: n = 31) WCG: 0.593 (pairs: n = 27)
Life satisfaction	19.44 (n = 34)	17.39 (n = 31)	0.183	22.06 (n = 34)	17.97 (n = 31)	0.012 *	IVG: 0.017 * (pairs: n = 34) WCG: 0.545 (pairs: n = 31)
General health status	3.38 (n = 34)	3.16 (n = 31)	0.504	3.29 (n = 34)	2.97 (n = 31)	0.170	IVG: 0.614 (pairs: n = 34) WCG: 0.175 (pairs: n = 31)
Depressive symptoms	49.21 (n = 33)	46.93 (n = 30)	0.730	61.06 (n = 34)	45.33 (n = 30)	0.015 *	IVG: 0.002 * (pairs: n = 33) WCG: 0.935 (pairs: n = 30)

* A *p*-value of less than 0.05 was considered statistically significant. Results are based on the information provided by participants interviewed at both time points, T0 and T1 (max. n = 65). IVG: intervention group; WCG: waiting control group. ^1^ Interview prior to the JOBS training (baseline). ^2^ Interview immediately after the JOBS training. ^3^ Outcome value change from T0 to T1.

### 3.6. Intervention Effect on Job Search-Specific Self-Efficacy and Health-Related Outcomes

Multivariate linear regression was applied to examine whether the treatment indicator variable was associated with the outcome variables. Table 4 and Table 5 report multivariate unstandardized regression coefficients, the coefficient of determination (R^2^), and the adjusted coefficient of determination (R^2^_adj._) for the three sequential models (M1 to M3) for each of the four outcomes. Furthermore, it can be determined whether there was a significant change in F from model to model.

In these regression analyses, the most important model for the main effects of the predictors is model 2, because it includes the relevant predictors (selected during different pre-analyses and based on theoretical considerations) but no interaction terms.

The interaction terms were added to model 3 for the examination of the hypothesized effect moderations (Hypothesis 5 and 6). However, in multiplicative interaction models (such as M3), the interpretation of the regression coefficients is often difficult and does not reflect the main effect of the single predictor. Rather, when interaction terms are included in the model, there are often situations in which the regression coefficients for each predictor variable say nothing about their actual effect on the outcome, and nor do their respective *p*-values [61,62].

**Hypothesis** **1**. Compared to the WCG, IVG reports higher levels of job search-specific self-efficacy.

On the item measuring job search-specific self-efficacy, where from 1 to 4 scale points could be scored, the level of job search-specific self-efficacy did not significantly differ at T0 (IVG: 2.72; WCG: 2.86 scale points). At T1, the IVG had reported an average of 0.32 points more than the WCG (*p* = 0.280). Although this is not a large difference, it is still noteworthy that the IVG had a significant, albeit small, increase from T0 to T1 (*p* = 0.049), while a non-significant decrease in job search-specific self-efficacy was found in the WCG (Table 3).

The results in the second row of Table 4 show that JOBS training had a significant positive effect on job search-specific self-efficacy in model 1 (*p* = 0.023 [one-sided]). The positive association remained in model 2, but the additional predictors included caused this positive association to fall just short of the required significance level (*p* = 0.052 [one-sided]).

Despite the loss of significance in model 2, there was a significant score change from T0 to T1 in the IVG. The consistent regression coefficients in model 1 and 2, coupled with the plausibility of the association (JOBS training is specifically designed to increase participants’ self-efficacy), support Hypothesis 1, which states that JOBS training has positive effects on job-search self-efficacy, especially considering the small sample size (lack of statistical power).

**Hypothesis** **2.**Compared to the WCG, IVG reports higher levels of life satisfaction.

On a scale from 5 to 35 points, the level of life satisfaction between IVG and WCG did not significantly differ at T0 (IVG: 19.44; WCG: 17.39 scale points; *p* = 0.183). In contrast, the difference at T1 (IVG: 22.06; WCG: 17.97 scale points) was clearly significant (*p* = 0.012). The increase from T0 to T1 in the IVG was relatively large and significant (+2.68 points; *p* = 0.017), while there was no relevant change in WCG (+0.29 points; *p* = 0.545) (Table 3).

The results displayed in the second row of Table 4 show that the JOBS training had a significant beneficial effect on life satisfaction in model 1. From model 1 to model 2, the positive regression coefficient remained relatively stable and statistically significant (*p* = 0.049). Compared to the WCG, the regression in M2 estimated that the IVG had a 2.736 scale point higher level of life satisfaction, clearly supporting Hypothesis 2.

In addition to the effect of the intervention, German citizenship was positively associated with life satisfaction.

**Hypothesis** **3.**compared to the WCG, IVG assesses its general health status more favorably.

On the scale from 1 to 5 points, the difference at T0 between both groups was negligible and statistically insignificant, with an average of 0.22 scale points more in the IVG compared to the WCG (*p* = 0.504). At T1, the difference was slightly larger (0.32), but still insignificant (*p* = 0.170), and the changes from T0 to T1 within both groups also showed no significant results (Table 3).

However, the results of the multivariate analysis (second row of Table 5) show that the JOBS training had a significant beneficial effect on the self-reported general health status in model 1 (*p* = 0.025). The positive regression coefficient of the treatment indicator variable remained stable from model 1 to model 2. The respective *p*-value in model 2 shows also statistical significance (*p* = 0.025 [one-sided]).

Although the descriptive results derived from Table 3 do not support a meaningful effect of the intervention, the stable estimates and significance in model 1 and model 2, as well as the plausible direction of the effect (JOBS training participants report better health as compared to WCG members) suggest an independent effect of the treatment indicator on the outcome variable. Compared to the WCG, the regression in M2 estimated that the IVG had a 0.337 scale point higher level of general health, clearly supporting Hypothesis 3.

**Hypothesis** **4.**Compared to the WCG, IVG reports lower levels of depressive symptoms.

On a scale from 0 to 100 points, the level of depressive symptoms between IVG and WCG at T0 did not significantly differ (IVG: 49.21; WCG 46.93 scale points; *p* = 0.730). However, the difference at T1 was clearly significant (*p* = 0.015) with 61.06 in the IVG and 45.33 scale points in the WCG. The increase from T0 to T1 in the IVG was large and significant (+10.56 scale points; *p* = 0.002), while there was no meaningful increase in the WCG (+1.89 points; *p* = 0.935) (Table 3).

The second row of Table 5 presents the results concerning self-reported mental health, assessed according to the WHO-5 Well-Being Index. The relatively large and stable regression coefficients clearly indicate that the JOBS training had a remarkable and significant beneficial effect on the mental health of the IVG participants in model 1 (*p* = 0.002) and 2 (*p* = 0.004).

The descriptive data of Table 3, the stable estimates and significance in model 1 and 2, as well as the plausible direction of the effect (JOBS training participants report better mental health compared to WCG members) suggest an independent effect of the treatment indicator on the outcome variable. Compared to the WCG, the regression in M2 indicated that the IVG had a 14.52 scale point higher level of mental health, clearly supporting Hypothesis 4.

**Hypothesis** **5 and 6.**The level of depressive symptoms and/or the duration of unemployment, respectively, before the JOBS training will moderate the effect of the intervention on mental health.

The results do not support the hypotheses that (1) the level of depressive symptoms and/or (2) the duration of unemployment before the JOBS training will moderate the effect of the intervention on mental health. First, the interaction terms included in model 3 have only small regression coefficients, and second, these estimates are far from statistically significant (“Intervention X Dur_Unempl”: *p* = 0.751; “Intervention × Depression”: *p* = 0.191).

**Table 4 ijerph-20-06835-t004:** Unstandardized coefficients resulting from OLS linear regression models for the effects of the JOBS Program Germany intervention and baseline predictors on job search-specific self-efficacy and satisfaction with life at first interview after the intervention (T1).

Independent Variables (Baseline)	Job Search-Specific Self-Efficacy (n = 54)			Life Satisfaction (n = 61)		
	M1	M2	M3	M1	M2	M3
Intervention ^1^ (yes vs. no)	0.449 *	0.338	0.344	2.774 *	2.736 *	7.743 *
Baseline control of outcome	0.525 ***	0.451 **	0.512 **	0.613 ***	0.493 ***	0.488 ***
Age		−0.014	−0.017		0.029	0.023
Gender (female vs. male) ^2^		−0.332	−0.351		−0.415	0.603
Education						
low vs. high		−0.768	−0.778		2.645	2.703
medium vs. high		−0.672	−0.642		2.865	3.071
Citizenship (yes vs. no)		0.488	0.560		4.544 *	4.930 *
Duration of unemployment ^3^		−0.004	−0.025		0.012	0.028
Depressive Symptoms		0.001	0.004		0.058	0.103 *
Interactions						
Intervention × Dur_Unempl ^4^			0.047			−0.110
Intervention × Depression ^5^			−0.007			−0.088
R^2^	0.314	0.449	0.473	0.408	0.510	0.531
adjusted R^2^	0.288	0.339	0.338	0.388	0.425	0.428
Significant change in F? (y/n) ^6^	y	n	n	y	n	n

In models 2 and 3, included predictors vary between the different outcome variables. They were selected (1) on a theoretical basis or (2) if they have been shown to be statistically associated with the outcome variable in different bi- and multivariate pretests at a significance level <0.2. * *p* < 0.05; ** *p* < 0.01; *** *p* < 0.001; A *p*-value of less than 0.05 was considered statistically significant. M1–M3: Model 1 to 3. R^2^: coefficient of determination. ^1^ Participation in the JOBS Training (IVG vs. WCG). ^2^ In the survey, the question for gender provided the following categories: “male”, “female”, “diverse” (no respondent selected “diverse”). ^3^ Duration of employment in years. ^4^ Interaction of the predictors “Intervention” multiplied by “Duration of unemployment” (“Dur_Unempl”). ^5^ Interaction of the predictors “Intervention” multiplied by “Depressive symptoms”. ^6^ shows whether R^2^ significantly increases from one to the next model specification.

**Table 5 ijerph-20-06835-t005:** Unstandardized coefficients resulting from OLS linear regression models for the effects of the JOBS Program Germany intervention and baseline predictors on mental health outcomes at first interview after the intervention (T1).

Independent Variables (Baseline)	General Health Status (n = 59)			Depressive Symptoms (n = 61)		
	M1	M2	M3	M1	M2	M3
Intervention ^1^ (yes vs. no)	0.378 *	0.337 *	0.847 *	15.073 **	14.524 **	30.341 *
Baseline control of outcome	0.709 ***	0.631 ***	0.593 ***	0.713 ***	0.704 ***	0.849 ***
Age		−0.007	−0.012		0.091	0.064
Gender (female vs. male) ^2^		−0.093	−0.016		2.592	5.744
Education						
low vs. high		−0.440	−0.468		4.591	4.732
medium vs. high		−0.106	−0.090		9.729	10.388
Citizenship (yes vs. no)		0.271	0.322		13.174	14.511
Duration of unemployment ^3^		−0.007	−0.021		−0.434	−0.424
Depressive Symptoms		0.001	0.008		― ^7^	― ^7^
Interactions						
Intervention × Dur_Unempl ^4^			0.021			−0.259
Intervention × Depression ^5^			−0.013			−0.291
R^2^	0.582	0.627	0.651	0.524	0.573	0.588
adjusted R^2^	0.568	0.559	0.571	0.508	0.509	0.508
Significant change in F? ^6^ (y/n)	y	n	n	y	n	n

In models 2 and 3, the included predictors vary between the different outcome variables. They were selected (1) on a theoretical basis or (2) if they have shown to be statistically associated with the outcome variable in different bi- and multivariate pretests at a significance level <0.2. * *p* < 0.05; ** *p* < 0.01; *** *p* < 0.001; A *p*-value of less than 0.05 was considered statistically significant. M1–M3: Model 1 to 3. R^2^: coefficient of determination. ^1^ Participation in the JOBS Training (IVG vs. WCG). ^2^ In the survey, the question for gender provided the following categories: “male”, “female”, “diverse” (No respondent selected “diverse”). ^3^ Duration of employment in years. ^4^ Interaction of the predictor “Intervention” multiplied by “Duration of unemployment” (“Dur_Unempl”). ^5^ Interaction of the predictor “Intervention” multiplied by “Depressive symptoms”. ^6^ shows whether R^2^ significantly increases from one to the next model specification. ^7^ For the outcome “Depressive symptoms” at T1, the regression coefficients for the predictor “Depressive symptoms” at T0 cannot be reported here because this predictor already corresponds to the predictor variable “Baseline control of outcome”.

## 4. Discussion

There are already a wide variety of interventions for the unemployed to promote health and reemployment [25,26,27,63]. One of those interventions is the JOBS Program evaluated in this study, which has been applied in many countries around the globe [30,32,34,35,36,37,38,39,40,41,64,65,66]. Before the pilot introduction in 2020, it had not yet been implemented in Germany on a larger scale. The introduction of the JOBS Program Germany as a nationwide pilot project allowed for the first systematic evaluation of this intervention in the German context.

The following hypotheses were examined.

Hypothesis 1 of this evaluation was that the JOBS Program Germany succeeded in increasing IVG participants’ job search-specific self-efficacy levels. The IVG significantly increased from T0 to T1. Multivariate linear regression model 1 yielded a significant association of the treatment indicator variable with the level of job search-specific self-efficacy while controlling for the baseline level of the outcome. This result is consistent with previous JOBS Program evaluation studies [31,32,39,41,67,68]. The fact that the underlying significance level for the association in model 2 was narrowly missed may also have been caused by the small number of included subjects (n = 54) that remained in the analysis.

Hypotheses 2 to 4 referred to three different dimensions of health-related outcomes, namely (1) life satisfaction, (2) general health, and—more specifically—(3) mental health. Despite the small sample, all three outcomes yielded statistically significant and meaningful positive associations with the participation in JOBS Program Germany.

To the best of our knowledge, “life satisfaction” has yet to be studied in JOBS Program evaluation studies. One comparable evaluation study by Proudfoot et al. [69] on the effect of cognitive-behavioral training on job-finding among long-term unemployed people also identified positive effects on life satisfaction. Moreover, our results seem conclusive since it is well known that unemployment negatively impacts many dimensions in the lives of those affected, as described in the introduction.

So far, the outcome “general health” has been examined only once within a JOBS Program evaluation study [38]. In this study, the authors reported neither meaningful and/or statistically significant effects nor what instrument they used. As explained above, in our study, we used the widely used item “How is your health in general” (see above) [53]. Semantically, it is a very broad question that basically covers all dimensions of health-related aspects, including somatic and mental health. It is therefore plausible that the JOBS Program Germany, which is a labor market and health-promoting intervention, has proven successful in having a positive effect on general health.

Concerning the positive results in terms of mental health outcomes, there are several JOBS Program evaluation studies internationally that are in line with our findings concerning the improvement of, e.g., depressive symptoms due to the JOBS training [31,34,35,36,42].

Hypothesis 5 and 6: The design of this study was oriented towards two previous studies from the United States and Finland [35,42]. Therefore, we strived to also examine moderating factors that were studied in those studies. We examined whether (1) the baseline level of depressive symptoms and/or (2) the duration of unemployment before the JOBS training would moderate the effect of the intervention on mental health. We therefore included additional interaction terms in our sequential regression model 3. In line with Vinokur et al. [42] and Vuori et al. [35], we did not identify moderating effects of the baseline level of depressive symptoms on the intervention effect in terms of mental health (depressive symptoms). Vinokur et al. [42] did not examine the moderating effects of the duration of unemployment and the results of Vuori et al. [35] showed no significant moderating effect of the duration of unemployment before the training. This is in line with our findings.

Previous studies in the USA and in Finland yielded positive effects. Therefore, we aimed to answer whether the JOBS Program will have similar effects for the unemployed in Germany, which has different labor markets, social security systems, and labor policies. However, if comparisons should be made, certain aspects that may influence the different study results should be considered. As we did not preselect a certain sample, e.g., based on specific demographic characteristics, not all study conditions could be replicated and therefore, our study differs from the Finnish study of Vuori et al. [35]. For example, at the time of recruitment, the respondents in Vuori et al.’s study [35] had a median age of 36 years (mean = 37.0 years; SD = 8.6), which is clearly younger compared to the German sample (median = 46 years; mean = 44.7 years; SD = 11.8). Also, the gender distribution varied between the two samples. Whereas in Vuori et al. [35], 77.8% were women and 22.2% men, these numbers were 45.6% (female) and 54.3% (male) in the German sample. Further, the Finnish sample had a median duration of unemployment of five months (mean = 10.7, SD = 17.3) and 28% were unemployed for 12 months or longer. In our sample, the median duration of unemployment was 48.5 months (mean = 76.7; SD = 73.3) and 88% were unemployed for 12 months or longer. There were also relevant differences between the German and the US-American sample of Vinokur et al. [42]. The participants of that sample were also much younger, with a median age of 34.7 years (mean = 36.2; SD = 10.38) and 45% of them were men and 55% women. Most of the participants had recently lost a job (mean = 4.11 weeks since job loss) and were unemployed for no longer than 13 weeks.

Also, the different time points of data collection after the intervention need to be considered when comparisons are made. The time points varied between the focused studies from the USA (Vinokur et al. [42]: two year follow-up) and Finland (Vuori et al. [35]: six months follow-up) as well as between our study, that report the data collection directly after the training but within four weeks.

Additionally, one needs to be aware of the different instruments used to collect data. To clarify, we can by and large successfully compare whether different studies have generally shown positive training effects, e.g., on depressive symptoms. However, it is not easy to directly compare the reported numbers of the different studies because we need to take into account the scale of measurement as well as the direction of the effect additionally to the “raw” numbers. Exemplarily, in Vinokur et al. (2000) [42], the training yielded a significant decrease in depressive symptoms of −0.06 scale points on a scale from 11 to 55 points. Vuori et al. (2002) [35] reported a non-significant decrease of −0.04 scale points on a scale from 0 to 30 points, and in our main model 2, we identified a significant (WHO-5 Well-Being Index) increase of +14.52 scale points on a scale from 0 to 100.

For a qualitative evaluation and the classification of the results, it is helpful to know the possible cut-off levels for the respective outcome and the level in the general population, possibly stratified for demographic characteristics like age groups.

Taking this into account, we consider our results as relevant. First, for the WHO-5 Well-Being Index, a representative sample of the German general population showed for the age group of 41–60 years (which matches the mean age of our sample best) a mean level of depressive symptoms of 69.95 scale points [70]. In contrast, our descriptive results indicate that the IVG had a mean level of depressive symptoms at T0 of 50.50 scale points (WCG: 43.44) (all participants at T0 included [n = 94]). This large difference between 50.50 and 69.95 scale points confirms the above mentioned findings that the unemployed on average suffer more frequently from mental health impairments than persons who are employed.

Second, this level of depressive symptoms in the IVG was only slightly above the cut-off score indicating screening for depression in clinical diagnosis (≤50 scale points; [55]) before the training, but was clearly higher and above the cut-off after the training with 61.06 scale points (WCG: 45.33). Additionally, the multivariate regression showed an estimated increase of 14.52 scale points (≈14.5%) as a result of the intervention, which underlines the relevance of this result because it is stated that (in individual diagnostics) a 10% difference (change) in the WHO-5 Well-Being Index should be considered relevant [71].

Comparing the results of our study with those of the example studies to be confirmed, our results appear clearer: Apart from statistical significance (which is indicated in our results) the reduction in depressive symptoms by −0.06 scale points on a scale from 11 to 55 points [42] and a reduction of −0.04 scale points on a scale from 0 to 30 points [35] seem to be less meaningful.

## 5. Limitations

This study originally aimed to examine the effects of the JOBS Program intervention on the reintegration into the labor market, life satisfaction, state of health, depressive symptoms, and psychological distress among the unemployed in Germany. Of secondary interest were moderating factors such as socio-demographic characteristics, the duration of unemployment, and the job-search intensity. The hampering conditions during the COVID-19 pandemic and its infection control measures caused a lack of participants, which led to a low statistical power. The authors are convinced that, under normal (non-pandemic) recruitment conditions, the statistical power would have produced clearer results, especially with regard to increasing self-efficacy expectations. With a larger sample, better subgroup analyses would have been possible; for example, with regard to the intervention effects among participants of different nationalities or regarding intervention effects in the different study centers.

## 6. Conclusions

This study contributes to existing literature on the effectiveness of the JOBS Program, which was studied in Germany for the first time. Shortly after the training sessions were conducted (T1), people who had participated in the JOBS training showed better life satisfaction, and better general as well as mental health. Although it was not statistically significant, job-search self-efficacy appeared to be positively associated with the JOBS training. Despite the enormous limitations created by the aftermath of the COVID-19 pandemic, especially with regard to recruiting participants, our results suggest that JOBS Germany is effective—at least in terms of health-related outcomes.

It is known that usually, mental health is more strongly affected among the long-term unemployed compared to short-term unemployed people [12] and our results suggest that the positive health effects of the JOBS Program do also apply to people who are long-term unemployed. In our study, 86% of the IVG was long-term unemployed (≥12 months). In addition, our sample was older than those in most other studies, which argues that JOBS training should also be offered to older unemployed workers who are looking for employment.

The introduction of the JOBS Program as a nationwide intervention in Germany could be another way for the target group to achieve a better future with better health and increased chances of reemployment. In addition to the current concept of face-to-face teaching, blended learning concepts could be used to replace or complement the current training program. This could increase the reach and would be more feasible for certain target groups who, for personal reasons, cannot easily complete a 20 h training course outside their home.

The results indicate that the training only partially increased the self-efficacy of the participants. Among other reasons, this could possibly also be related to the fact that the participants in the observation period were aware that reintegration into the labor market was made even more difficult due to the COVID-19-related lockdowns [72].

However, since self-efficacy expectations are also based on personal experience of positive coping, it could also be helpful to offer individual counseling after the training sessions. In this way, the trainers could support the participants individually in the application process beyond the training units; for example, in the preparation of contacts with employers or regarding job interviews. Such “milestones” could be followed up and the positive aspects of these activities could be worked out—similar to the JOBS training sessions.

The results obtained under the difficult conditions suggest that the further development of the JOBS Program Germany is promising. We therefore recommend that, if the JOBS Program intervention is offered to unemployed people in Germany in the future, its effectiveness should be further investigated to verify the results obtained here and to allow the examination of further endpoints, effect moderations, and subgroups.

It should also be investigated to what extent the concept of the JOBS Program can be used even for groups beyond the unemployed. This is because the training is primarily based on strengthening personal resources such as self-efficacy expectations, self-esteem, self-confidence, etc., and can therefore certainly be extended to other, e.g., socially disadvantaged, groups of people.

## Figures and Tables

**Figure 1 ijerph-20-06835-f001:**
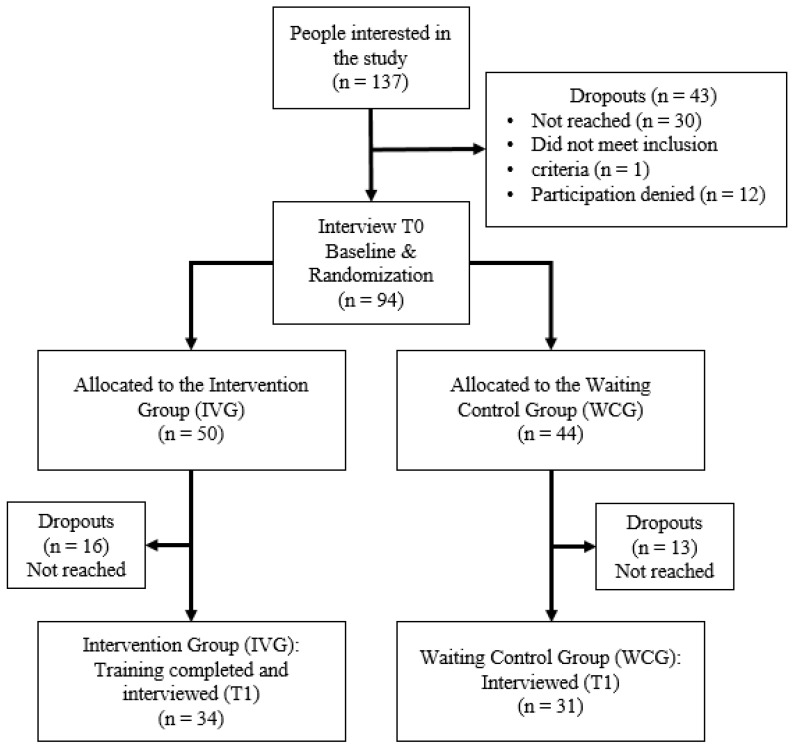
Flow chart of participant recruitment.

## Data Availability

The data presented in this study are not available from the corresponding author. The data used for this work were collected from volunteer participants who consented to the data collection and the corresponding data protection concept. This data protection concept includes that data will not be shared with others not involved in this research project and that data collected from subjects will be deleted 5 years after completion of data collection or no later than 31 December 2028. The data protection concept was reviewed and approved by the departments responsible in the individual institutions. All data were collected, transmitted, stored, and deleted in strict compliance with the German Federal Data Protection Act and the European General Data Protection Regulation (DSGVO).

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
