# Peer review of "Results from a Nationwide Evaluation Study of Labor Market-Integrative Health Promotion for the Unemployed: Impact of the JOBS Program Germany"

_ijerph, 2023, doi:10.3390/ijerph20196835_

Round 1
Reviewer 1 Report
I found this article an interesting read, well written with clearly understood analysis.
My only comments for improvement centre on the review of the literature and the limitations, further recommendations.
I felt more literature could be referred to, particularly more recent literature. I acknowledge some recent literature has been referred to but given the range of academic research in the area particularly since 2019, I would have expected more recent literature to be referenced.
The limitations and further recommendations, although included could have been expanded to underline the contribution and assist research moving forward.
Good work here, thank you for sharing.
I have no issue with the quality of English used in the paper.
Author Response
1.) Minor editing of English language required:
Authors' response: We thank the reviewer for the advice. We will use the MDPI Language Editing Service.
2.) I felt more literature could be referred to, particularly more recent literature. I acknowledge some recent literature has been referred to but given the range of academic research in the area particularly since 2019, I would have expected more recent literature to be referenced:
Authors' response: We thank the reviewer for this comment. We introduced more recent literature and added the aspect „substance-use disorders” among the unemployed, also with recent literature reference: Nolte-Troha, C.; Roser, P.; Henkel, D.; Scherbaum, N.; Koller, G.; Franke, A.G. Unemployment and Substance Use: An Updated Review of Studies from North America and Europe. Healthcare 2023, 11, 1182, doi:10.3390/healthcare11081182.
3.) The limitations and further recommendations, although included could have been expanded to underline the contribution and assist research moving forward:
Authors' response: We thank the reviewer for this recommendation. Concerning the limitations, we complemented the text with the following part: “The authors are convinced that, under normal (non-pandemic) recruitment conditions, the statistical power would have produced clearer results, especially with regard to increasing self-efficacy expectations. With a larger sample, better subgroup analyzes would have been possible, for example with regard to the intervention effects among participants of different nationalities or with regard to the intervention effects in the different study centers“.
Concerning the recommendations with respect to “research moving forward” (that should be part of the conclusions in our view), we added the following part: “We therefore recommend that, if the JOBS Program intervention is offered to unemployed people in Germany in the future, its effectiveness should be further investigated in order to verify the results obtained here and to be able to examine further endpoints, effect moderations and subgroups”.
Reviewer 2 Report
The topic of the paper is interesting. Some suggestions for the paper are:
-. Please also write the originality and contribution(s) of this study in Introduction, not only in Conclusions.
-. Please add the research model in the manuscript.
The English quality of the manuscript is good. However, I suggest the authors to use the simple present tense in explaining the current study. For example, "We applied a randomized controlled trial..." in Abstract should be written as "We apply a randomized controlled trial..."
Author Response
1.) Please also write the originality and contribution(s) of this study in Introduction, not only in Conclusions.:
Authors' response: We thank the reviewer for this suggestion. By the statements in the introduction on the confirmatory character of the study (namely the evaluation of the effects of the JOBS Program, which was done so comprehensively for the first time in Germany) we intended to point out the originality and the contribution of the study. We have now clarified this somewhat with the inserted phrase "[...] for the first time [...]". For clarification, the whole sentence: "This confirmatory study for the first time examined the effects of the JOBS Program on the job-search specific self-efficacy and on health-related outcomes such as life satisfaction and depressive symptoms among the unemployed in Germany."
2.) Please add the research model in the manuscript.:
Authors' response: We thank the reviewer for the suggestion. However, because this is a confirmatory evaluation study, the research model and quantitative empirical methods follow the original studies, i.e., the studies by Vinokur et al. (2000) and Vuori et al. (2002) (see introduction). Nevertheless, your comment prompted us to add a paragraph that explains - based on three relevant examples of selected theories/models - how unemployment and mental health are associated.
3.) The English quality of the manuscript is good. However, I suggest the authors to use the simple present tense in explaining the current study [...]:
Authors' response: We thank the reviewer for the advice. We will use the MDPI Language Editing Service.
Reviewer 3 Report
First and foremost, thank you for providing me with the opportunity to read this article. The article is generally well-written and straightforward. However, I do have a few ideas. With these suggestions, I believe the study will be more significant.
Comments to the Author
The background research appears to be inadequate. The link between mental health and unemployment has yet to be fully explained. It was stated that the JOBS program had a beneficial impact in other nations where it was adopted, such as the United States and Finland. However, no mention was made of the intervention program's differential effects on age, gender, education, unemployment duration, race, ethnicity, or groups with special needs. It would be excellent to expand on this part a little.
Germany's uniqueness is not mentioned. It is not explained why elderly and long-term unemployed people are in the majority. What are the names of the five non-citizens in the research group? A background understanding of Germany appears to be required, such as unemployment, unemployed groups, and some basic labor market data.
Because the conclusion section repeats the findings, its original contribution to the literature is unclear. Rather than summarize the findings, write a conclusion highlighting the key points.
no comment
Author Response
1.) The background research appears to be inadequate. The link between mental health and unemployment has yet to be fully explained. It was stated that the JOBS program had a beneficial impact in other nations where it was adopted, such as the United States and Finland. However, no mention was made of the intervention program's differential effects on age, gender, education, unemployment duration, race, ethnicity, or groups with special needs. It would be excellent to expand on this part a little:
Authors' response: Thanks for this comment. (i) We introduced a section that refers to the fact that the link between unemployment and mental health is empirically well proven and explains - based on three relevant examples of selected theories/models - how this link can be explained. (ii) Most of the subgroup analyzes mentioned by the reviewer were not carried out in the studies that were strived to confirm with our study. However, these studies examined effect modifications by the duration of unemployment, by job search intensity and by the baseline level of depressive symptoms. Our study also sought to investigate these moderating effects (see hypotheses and results). Subgroup analyzes could not be carried out in our study because the sample was too small. We have now added a text to this in the limitations and conclusions.
2.) Germany's uniqueness is not mentioned. It is not explained why elderly and long-term unemployed people are in the majority. What are the names of the five non-citizens in the research group? A background understanding of Germany appears to be required, such as unemployment, unemployed groups, and some basic labor market data:
Authors' response: Thanks for this suggestion. This is already a lengthy manuscript and the main aim of this study was the evaluation of the JOBS Program intervention in Germany. Therefore, we introduced only a short paragraph concerning the number of unemployed people and of the long-term unemployed as well as the rates. We hope that is acceptable for the reviewer.
We are sorry, we do not understand the third sentence (The question "What are the names of the five non-citizens in the research group?").
3.) Because the conclusion section repeats the findings, its original contribution to the literature is unclear. Rather than summarize the findings, write a conclusion highlighting the key points:
Authors' response: We are sorry, we think that we did not just repeat/summarize the findings in the conclusion section. Rather we did in deed highlight the original contribution of this study and discussed the findings in the context of the relevant literature on the JOBS Program interventions in other countries. This follows the confirmative nature of this study. Nevertheless this comment prompted us to introduce some new literature and to more emphasize the original contribution of our study.
Round 2
Reviewer 2 Report
Thank you for revising the manuscript. Please revise the part of "Error! Reference source not found" in the manuscript.
Please make sure that the MDPI Language Editing Service is used before submitting the final version of the paper.
Author Response
We again thank the reviewer for the suggestions! Please find below our responses:
1.) ("Please revise the part of "Error! Reference source not found" in the manuscript."):
We do not know how this could happen, but we will again insert the in-text citations and check the reference list accordingly.
2.) ("Please make sure that the MDPI Language Editing Service is used before submitting the final version of the paper."):
We will care about the English editing and we are contact with the IJERPH editors concerning this issue.
With many thanks and kind regards
Heiko